**communications** sustainability

# Realistic AI-generated climate disaster images decrease support for climate action when artificial origin is suspected
Fabienne Bünzli ⬤ ✉ & Martin J. Eppler ⬤

Policymakers and environmental advocacy organizations are increasingly using AI-generated images of climate disasters to advocate for climate interventions. Here we show that highly realistic AI-generated climate disaster images do not increase support for climate action. Three large-scale experiments ($N = 2580$) provide evidence that these images intensify emotional responses but also elicit resistance in the form of reactance and reduced trust in the message source. These adverse effects are especially pronounced for highly realistic images that are suspected to be AI-generated or explicitly labeled as such. Moreover, when their AI origin is suspected but not disclosed, such images significantly reduce individuals' willingness to make personal sacrifices for climate action. Overall, our findings call for an informed use of generative AI in climate advocacy that accounts for unintended effects and challenges the assumption that highly realistic AI-generated disaster images are inherently persuasive.

Although a majority of Americans report being aware of the risks of climate change, many remain reluctant to support climate interventions that entail personal sacrifices, such as higher gasoline taxes, increased meat prices, or stricter energy regulations[1-3]. Consequently, policymakers and advocacy organizations often face resistance when promoting such interventions[4].

One common strategy to overcome this resistance is to underscore the urgency of climate action by highlighting catastrophic consequences of climate change, such as rising sea levels and extreme weather events like wildfires and hurricanes[5-8]. With recent advances in generative artificial intelligence (GenAI), a growing number of initiatives have begun harnessing AI technologies to produce visual depictions of climate disasters[9]. One example is the *FloodVision* initiative, which gathers visual data from coastal communities in the US and uses generative AI to create photorealistic visualizations of flooding and sea-level rise[10]. Such initiatives are designed to help audiences imagine what climate consequences might look like in familiar places, thereby making climate change more tangible, concrete, and less psychologically distant[11,12]. In Europe, one project invited participants to use generative AI to visualize climate disaster scenarios affecting places of personal significance[12]. In Panama, an initiative by the United Nations Development Programme (UNDP) encouraged students to experiment with generative AI to depict how floods, droughts, or earthquakes might devastate their own neighborhoods[13].

Beyond such individually tailored, co-creative initiatives, AI-generated imagery has become increasingly common in mass-mediated advocacy campaigns. For example, Greenpeace Philippines published a series of AI-generated images for Earth Day during the Plastic-Free Future Youth Forum, depicting young people's visions of a plastic-free future[14]. Similarly,

WWF UK launched a *Future of Nature* exhibition using generative AI to visualize the bleak consequences for ecosystems if humanity fails to take decisive action against climate change[15]. Here, we focus on *AI-generated climate disaster images used in mass-mediated advocacy campaigning*. We show that highly realistic images overall fail to yield a persuasive advantage and can even backfire when their AI origin is suspected but not disclosed, resulting in reduced willingness to make personal sacrifices for climate action. Importantly, the highly realistic images used in this research approached photorealism, while still retaining stylized features that were characteristic of AI-generated imagery at the time of data collection (e.g., dramatic lighting).

Generative AI encompasses a variety of models capable of generating images by learning patterns from large datasets of human-generated visual content[11]. Popular generative AI tools include ChatGPT, Midjourney, and Stable Diffusion. Their growing use for climate disaster visualizations is driven by several factors, including scalability, adaptability, and the ability to depict hypothetical futures from simple textual prompts[11,12,16].

Most fundamentally, AI-generated climate disaster images can be distinguished by their visual realism, defined here as the extent to which an image preserves perceptual cues of physical reality[16,17]. Images low in visual realism rely on schematic, illustrative, or abstract representations, including icons and symbolic graphics. They are typically characterized by simplified forms, flattened or uniform color fields, reduced surface texture, limited or absent depth cues, and non-naturalistic lighting. By contrast, images high in visual realism convey more detailed, lifelike depictions of physical scenes. They are commonly characterized by fine-grained surface texture, naturalistic lighting and shading, atmospheric effects (e.g., smoke or haze), and

---

Institute for Media and Communications Management, University of St.Gallen, St.Gallen, Switzerland. ✉e-mail: fabienne.buenzli@unisg.ch

pronounced spatial depth, thereby approximating the appearance of photographs. Note that visual realism is distinct from stylization, which refers to the extent to which an image applies artistic transformations to its representation, for example, through filters, dramatic color grading, or exaggerated contrast. Importantly, stylization can occur at both low and high levels of visual realism.

The capacity of a generative AI system to produce images with high visual realism is commonly treated as a benchmark of its maturity[18]. Rapid technological advances in recent years have substantially narrowed the gap between AI-generated images and photographs, making it increasingly difficult for audiences to reliably distinguish between the two[19]. For instance, a recent study demonstrated that people are often unable to differentiate AI-generated human faces from real photographs[20].

This development gives rise to a second, conceptually important dimension: perceived origin, which refers to whether audiences identify a highly realistic AI image as a photograph or as an artificially generated depiction[21]. One approach to reducing ambiguity about an image's origin and enhancing transparency is to add a verbal label to AI-generated images[11,22]. The present research examines the extent to which audience responses to AI-generated climate disaster images are shaped by the visual realism of such images as well as by their perceived origin.

Disaster images portray the harmful consequences of climate change and thus depict threats to human lives and human civilization at large[23]. In terms of processing, such visual threat depictions have been associated with increased emotional responses[24,25]. On the one hand, disaster imagery may foster harm-oriented emotional processing, in which viewers focus on the depicted damage and experience corresponding negative emotions such as fear, sadness, despair, and guilt[26–29]. Climate advocacy messages commonly seek to elicit such emotional responses, as these emotions are known to enhance persuasion[30,31]. For example, Wong-Parodi and Feygina[24] show that experiencing negative emotions increases acceptance of climate change, concern about climate risks, and willingness to take climate action.

We argue that AI-generated disaster images heighten harm-oriented emotional processing. These effects should be particularly pronounced for highly realistic images, as their lifelike depiction of climate threats makes harmful consequences more salient and tangible (H1)[17]. Moreover, harm-oriented emotions are expected to increase individuals' willingness to make personal sacrifices for climate action (H2)[24].

Persuasive messages can elicit multiple, sometimes competing, responses[32,33]. In addition to harm-oriented processing, climate disaster images may also trigger resistance-oriented processing[7,34,35]. This form of processing is characterized by heightened awareness of the message source's persuasive intent and by negative emotions, including anger. One common manifestation of resistance-oriented processing is psychological reactance, an amalgam of anger and counter-arguing that prompts individuals to push back against (climate) advocacy[36,37]. Importantly, reactance arises when people feel that a message is manipulative and constrains their personal freedom[36–38]. Emerging research from environmental and health contexts suggests that certain visual images can increase perceptions of threat to freedom, which in turn trigger reactance and reduce message compliance[35,39,40].

Disaster images may elicit psychological reactance when recipients perceive dramatic scenes as attempts to constrain their freedom by pressuring them to engage in climate action[7,34]. We expect perceived threat to freedom to be especially pronounced for highly realistic AI-generated disaster images, as their vivid, lifelike depiction of damage makes climate consequences feel more immediate and experiential, thereby increasing perceived pressure to take climate action (H3). Moreover, perceived threat to freedom may be further amplified when audiences suspect or know that an image is AI-generated, as such awareness may make the imagery appear deliberately engineered to exert persuasive pressure (H4). Finally, perceived threat to freedom is expected to elicit psychological reactance, which in turn should be associated with reduced willingness to make personal sacrifices for climate action (H5).

In addition to psychological reactance, resistance-oriented processing may also manifest through reduced perceived trustworthiness of the message sender[22,41]. Trustworthiness refers to the extent to which a communicator is perceived as credible, reliable, and acting in good faith, and is known to enhance message persuasiveness[42]. Critically, AI-generated imagery has been associated with perceptions of artificiality and inauthenticity, as well as diminished trust[11,43]. Accordingly, we propose that an image's perceived origin shapes individuals' responses to disaster visualizations. When audiences suspect that an image may be AI-generated (e.g., because it appears artificial) or are informed of its AI origin through an accompanying text label, they may perceive the message sender as less trustworthy (H6). Because perceived trustworthiness is positively associated with persuasion, declines in trust are likely to undermine advocacy efforts and reduce individuals' willingness to make personal sacrifices for climate action (H7).

To test our hypotheses, we conducted three consecutive studies. The first study examined the effects of visual realism in AI-generated climate disaster images. Using Midjourney's text-to-image generative AI tool, we created visualizations of severely flooded cities. Participants were randomly assigned to one of three conditions (see Fig. 1): (1) a text-only control condition featuring a simple call for climate action; (2) a low visual realism condition combining the same call with an AI-generated image depicting the flooded city in a schematic and illustrative format, characterized by abstracted forms, reduced texture, and limited depth cues; or (3) a high visual realism condition featuring the same call paired with an AI-generated image approximating photographic appearance through fine-grained detail, naturalistic lighting, and spatial depth. Although the images reflected state-of-the-art generation capabilities at the time of data collection, their detailed yet somewhat stylized aesthetic retained features characteristic of earlier AI-generated imagery. By contemporary standards, they may therefore not appear fully photorealistic.

Study 2 replicated the previous study using a different set of visual stimuli. The images were created using text-to-image generation in Midjourney and depicted fictitious cities resembling those in U.S. states that are particularly vulnerable to wildfires (California), hurricanes (Louisiana), and coastal flooding (Florida). Participants were again randomly assigned to one of three conditions: (1) a text-only condition; (2) a low visual realism condition; or (3) a high visual realism condition. Importantly, the high-realism images in this study also retained stylistic features characteristic of AI-generated imagery at the time of data collection.

Study 3 extended the previous studies by examining the role of the perceived origin of highly realistic AI-generated climate disaster images. For the sake of consistency and comparability, we used the same high–visual realism images as in Study 2. Participants were randomly assigned to one of three conditions: (1) a text-only message containing a call to action; (2) the same message accompanied by a highly realistic AI-generated climate disaster image; and (3) the same message and image accompanied by an additional AI-origin label stating that the image was "created by artificial intelligence" (see Appendix, Fig. A.3). Participants in the image conditions (2 and 3) were asked whether they believed the image was a photograph, AI-generated, or whether they were unsure about its origin.

## Results

The final sample for Study 1 included 534 participants. All measures and items are provided in the Appendix (Table A.5). To examine the hypothesized competing processing mechanisms—harm-oriented versus resistance-oriented processing—we employed structural equation modeling (see Fig. 2). A confirmatory factor analysis was conducted to validate the measurement model prior to estimating the structural equation model (Fig. A.4). The structural equation model included two exogenous dummy-coded variables: one indicating the presence of a low visual realism AI image (1 = present, 0 = absent), and another indicating the presence of a high visual realism AI image (1 = present, 0 = absent). For more information on the coding of the exogenous variables, see Tables A.9–A.11 in the Appendix. Harm-oriented emotions, perceived threat to freedom, psychological reactance, and willingness to make personal sacrifices for climate action were included as endogenous variables. The exogenous variables were allowed to correlate with one another, but residual covariances among the

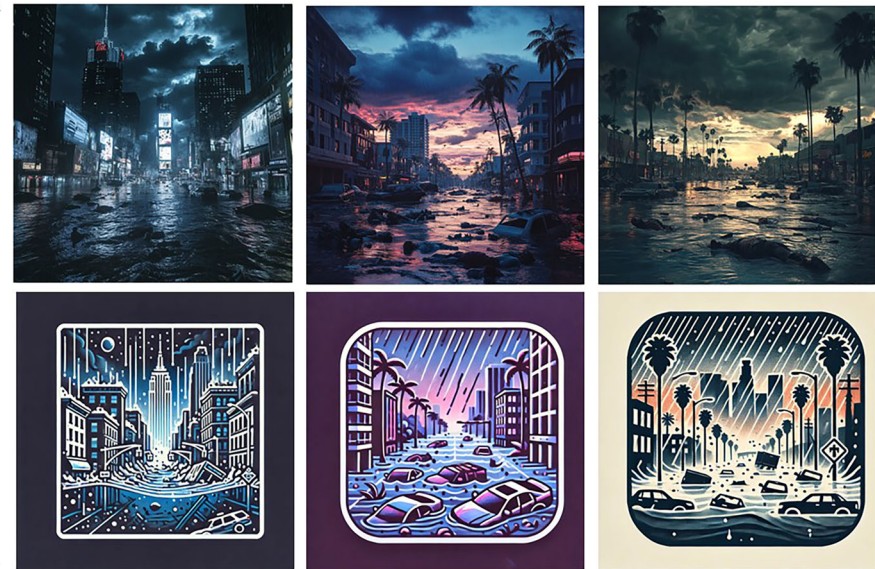

Visual realism of AI-generated climate disaster images

**Fig. 1 |** Visual realism in AI-generated climate disaster images increases from bottom to top, ranging from low to high. Here, we show that highly realistic AI-generated depictions of severe climate impacts can undermine support for climate action when their artificial origin is suspected.

endogenous variables were not permitted. Age, gender, political orientation, and prior disaster experience (1 = yes, 0 = no or don't know) were added as control variables. The model demonstrated a good fit to the data.

The presence (vs. absence) of AI-generated disaster images both high and low in visual realism increased harm-oriented emotions, including fear, sadness, guilt, and despair (Correlation table is provided in the Appendix, Table A.6). High visual realism images had a considerably greater effect ($b = 1.05$, SE = 0.17, $p < 0.001$) on these emotional responses than low visual realism images ($b = 0.69$, SE = 0.17, $p < 0.001$). These findings support H1. In line with H2, harm-oriented emotions positively predicted individuals' willingness to make personal sacrifices for climate action ($b = 0.43$, SE = 0.05, $p < 0.001$).

At the same time, high visual realism images yielded a significant effect on perceived threat to freedom ($b = 0.83$, SE = 0.16, $p < 0.001$), while low visual realism images did not ($b = 0.07$, SE = 0.16, $p = 0.673$). These findings align with H3. Perception of threat to freedom significantly increased psychological reactance ($b = 0.54$, SE = 0.05, $p < 0.001$), which in turn was associated with lower willingness to make personal sacrifices for climate action ($b = -0.70$, SE = 0.09, $p < 0.001$). Thus, the data support H5.

Finally, we conducted an analysis of covariance to examine the total effects of visual realism on willingness to make personal sacrifices for climate action. Age, gender, political orientation, and prior disaster experience served as control variables (see Table A.12 for the corresponding analysis of variance without control variables). The analysis revealed a significant main effect of experimental condition on willingness to make personal sacrifices, $F(2, 527) = 3.39$, $p = 0.035$, $\eta^2_p = 0.013$. Participants in the low visual realism AI condition ($M = 4.6$, SD = 1.6) reported a higher willingness to make personal sacrifices than those in the text-only control condition ($M = 4.2$, SD = 1.6; $p = 0.059$). This difference was statistically significant in the model without control variables ($p = 0.044$), but exceeded the conventional significance threshold of .05 when control variables were included ($p = 0.059$). Although participants reported a slightly higher willingness to make personal sacrifices when high visual realism AI imagery was present compared to the text-only condition ($M = 4.5$, $SD = 1.7$), this difference was not statistically significant either in the model without control variables ($p = 0.208$) or in the model with control variables ($p = 0.093$).

The final sample for Study 2 encompassed 552 participants. The study was designed to replicate the effects of visual realism in AI-generated disaster images. We again used the same structural equation modeling approach, including age, gender, political orientation, and prior disaster

experience as control variables (Fig. 3; see Fig. A.5 for the confirmatory factor analysis). In line with Study 1, exposure to both high- and low-visual-realism AI-generated disaster images increased harm-oriented emotions, including fear, sadness, guilt, and despair (Correlation table is provided in the Appendix, Table A.7)[26-29]. Moreover, high visual realism images again yielded a considerably stronger effect ($b = 0.93$, SE = 0.17, $p < 0.001$) on these emotional responses than low visual realism images ($b = 0.38$, SE = 0.16, $p = 0.020$). These findings provide further support for H1. Consistent with H2, harm-oriented emotions again positively predicted individuals' willingness to make personal sacrifices for climate action ($b = 0.38$, SE = 0.04, $p < 0.001$).

Consistent with Study 1, high visual realism images significantly increased perceived threat to freedom ($b = 0.62$, SE = 0.14, $p < 0.001$). In Study 2, low visual realism images also produced a significant effect on perceived threat to freedom ($b = 0.42$, SE = 0.14, $p = 0.004$). Importantly, the effect of high visual realism was considerably stronger than that of low visual realism, providing support for H3. Perception of threat to freedom again induced reactance ($b = 0.31$, SE = 0.04, $p < 0.001$), which in turn was correlated with lower willingness to accept personal sacrifices for climate action ($b = -1.05$, SE = 0.18, $p < 0.001$). Thus, the data support H5.

A subsequent analysis of covariance revealed no significant effect of visual realism on willingness to make personal sacrifices for climate action, $F(2, 545) = 1.862$, $p = 0.156$, $\eta^2_p = 0.007$. The non-significant effect can be explained by the concurrent activation of both harm-oriented and resistance-oriented processing.

Overall, across both studies, high-visual-realism images showed a consistent pattern: they elicited two competing processing mechanisms that offset one another. As a result, the inclusion of highly realistic AI-generated disaster imagery did not produce a persuasive advantage.

The final sample for Study 3 encompassed 1494 participants. The study was designed to further validate findings on visual realism while also examining the role of perceived image origin. For the sake of consistency and comparability, we used the same high–visual realism images as in Study 2. Participants were randomly assigned to one of three conditions: (1) a text-only message; (2) the same message accompanied by a highly realistic AI-generated climate disaster image; or (3) the same message and image accompanied by an AI-origin label stating that the image was "created by artificial intelligence". Within the second condition, we further distinguished whether participants believed the image was a real photograph or suspected it was AI-generated.

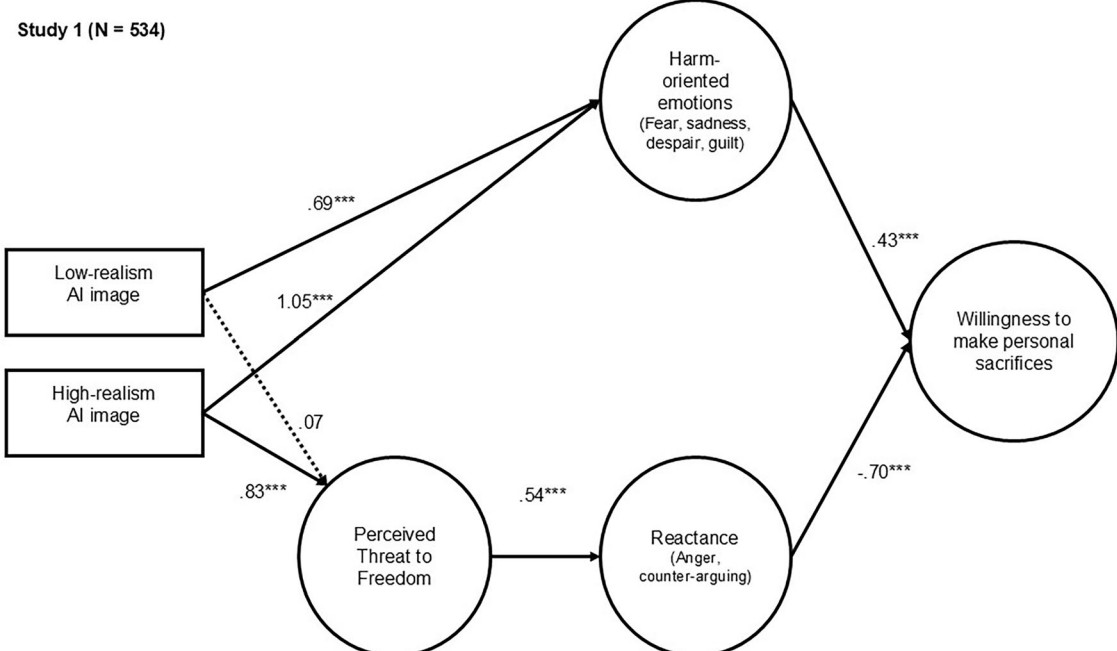

**Fig. 2 | Study 1: Structural model showing how visual realism in AI-generated climate disaster imagery influences individuals' willingness to make personal sacrifices for climate action.** Path coefficients are unstandardized weights. Age, gender, political orientation, and prior disaster experience were included as control variables. Dashed lines indicate nonsignificant paths. $*p \leq 0.05$. $**p \leq 0.01$. $***p \leq 0.001$. Model fit Study 1: $\chi^2(260, N = 534) = 830.585, p < 0.001$; RMSEA = 0.064 (90% confidence interval [CI] = 0.059–0.069); CFI = 0.936; and TLI = 0.919.

To test our hypotheses, we used structural equation modeling (Fig. 4; see Fig. A.6 for the confirmatory factor analysis). The model included three exogenous dummy-coded variables that distinguished between participants who saw an AI image and believed it was a photograph (1 = present, 0 = absent), participants who saw an AI image and believed it was AI-generated or were unsure about its origin (1 = present, 0 = absent), and participants who saw an AI image accompanied by an explicit AI-origin label (1 = present, 0 = absent). Harm-oriented emotions, perceived threat to freedom, psychological reactance, perceived trustworthiness, and willingness to make personal sacrifices for climate action were included as endogenous variables. The exogenous variables were allowed to correlate with one another, but residual covariances among the endogenous variables were not permitted. Age, gender, and political orientation were added as control variables. The overall model fit was good.

The presence of a highly realistic climate disaster image enhanced harm-oriented emotions across all types of origin-perceptions (Perceived as photo: $b = 1.07$, SE = 0.14, $p < 0.001$; Perceived as AI-generated or unsure: $b = 1.10$, SE = 0.10, $p < 0.001$; AI cue present: $b = 1.11$, SE = 0.10, $p < 0.001$) (Correlation table is provided in the Appendix, Table A.8). These findings provide further support for the capacity of highly realistic AI-generated disaster imagery to elicit harm-oriented emotional processing. As predicted by H2, these emotions were again positively associated with individuals' willingness to make personal sacrifices for climate action ($b = 0.32$, SE = 0.03, $p < 0.001$).

However, differences in perceived origin emerged for resistance-oriented processing. While all images significantly increased perceived threat to freedom, the effects were roughly twice as large when images were suspected to be AI-generated or accompanied by an explicit AI cue (Perceived as photo: $b = 0.33$, SE = 0.13, $p = 0.012$; Perceived as AI-generated or unsure: $b = 0.68$, SE = 0.10, $p < 0.001$; AI cue present: $b = 0.60$, SE = 0.09, $p < 0.001$). These findings are consistent with H4. Perceived freedom threat, in turn, increased psychological reactance ($b = 0.52$, SE = 0.03, $p < 0.001$), which was associated with reduced willingness to make personal sacrifices for climate action ($b = -0.68$, SE = 0.06, $p < 0.001$), providing further support for H5.

In addition, both suspicion of AI generation and explicit labeling, but not perceptions of the image as a photograph, reduced the perceived trustworthiness of the message source (Perceived as AI-generated or unsure: $b = -0.62$, SE = 0.13, $p < 0.001$; AI origin label present: $b = -0.60$, SE = 0.12, $p < 0.001$). These findings support H6. Trustworthiness, as anticipated in H7, was positively associated with willingness to make personal sacrifices for climate action ($b = 0.27$, SE = 0.02, $p < 0.001$).

Results of an analysis of covariance revealed significant differences in willingness to make personal sacrifices for climate action across conditions, $F(3, 1487) = 8.820, p < 0.001, \eta^2_p = 0.017$ (Fig. 5). The findings showed that participants were significantly *less* willing to make personal sacrifices for climate action when they believed the high visual realism image was AI-generated ($M = 4.3$, SD = 1.8), compared to when they believed it was a photograph ($M = 4.9$, SD = 1.6, $p = 0.001$) or saw the text-only message ($M = 4.8$, SD = 1.6, $p < 0.001$). Participants who viewed images accompanied by an AI origin label ($M = 4.6$, SD = 1.6) reported lower willingness to make personal sacrifices than those who believed the image was a photograph or viewed the text-only message. However, these differences did not reach statistical significance ($p = 0.051$ and $p = 0.101$, respectively).

Taken together, these findings replicate the effects observed in Studies 1 and 2: high *visual realism* in AI-generated climate disaster images evokes strong emotional responses, but these also include reactance. Our research further demonstrates that perceived image origin matters. Highly realistic climate disaster images that are believed to be AI-generated or explicitly labeled as such elicit heightened reactance and reduce trust in the message sender, both of which are negatively associated with willingness to make personal sacrifices for climate action. Moreover, when an image's AI origin is suspected but not transparently disclosed, message effectiveness significantly diminishes. Overall, our research serves as a cautionary note for climate advocacy. Although intended to enhance persuasion, such imagery can inadvertently undermine support for climate action.

## Discussion

Generative AI has been widely described as a potential game changer for mass-mediated climate advocacy[11,12]. Given its ability to visualize virtually

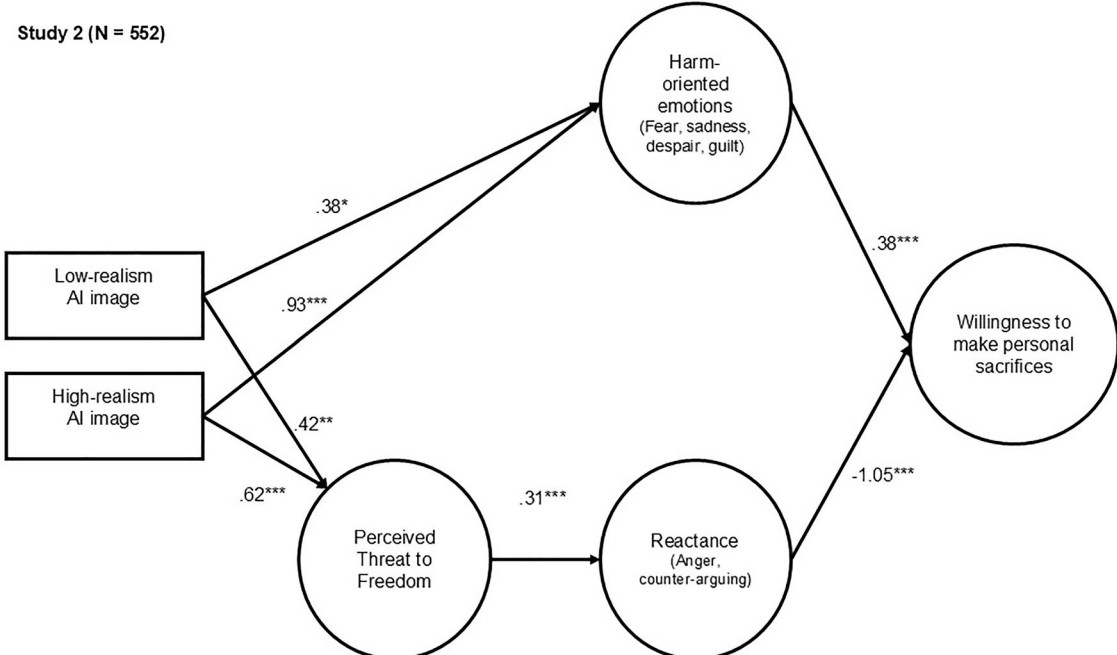

**Fig. 3 | Study 2: Structural model showing how visual realism in AI-generated climate disaster imagery influences individuals' willingness to make personal sacrifices for climate action.** Path coefficients are unstandardized weights. Age, gender, political orientation, and prior disaster experience were included as control variables. Dashed lines indicate nonsignificant paths. $*p \leq 0.05$. $**p \leq 0.01$. $***p \leq 0.001$. Model fit Study 2: $\chi^2(260, N = 552) = 972.142$, $p < 0.001$; RMSEA = 0.071 (90% confidence interval [CI] = 0.066–0.075); CFI = 0.927; and TLI = 0.909.

any scenario quickly and at scale, it is unsurprising that policymakers, nonprofit organizations, and climate actors have increasingly begun to employ AI-generated climate disaster imagery. However, our findings urge caution when using such visuals, particularly when they are highly realistic. Evidence from three experiments suggests that these images activate competing psychological responses that offset one another. On the one hand, highly realistic AI-generated disaster images amplify emotional responses to the disasters shown, including fear, sadness, despair, and guilt. On the other hand, they elicit resistance toward the communicator's persuasive intent. Participants reported feeling more manipulated and more threatened in their autonomy when exposed to such imagery, which in turn increased psychological reactance, a motivational state characterized by anger and counterarguing. Notably, these effects were more pronounced and consistent for AI-generated disaster images with high visual realism compared to low visual realism.

Furthermore, when highly realistic images were suspected or labeled as artificially generated, reactance was amplified relative to when the same images were believed to be authentic photographs. In addition, our findings show that when disaster images are believed to be AI-generated or labeled as such, rather than believed to be authentic photographs, they reduce the perceived trustworthiness of the message sender. While our findings support the notion that climate disaster images can elicit strong emotional engagement, they also demonstrate that such responses are not uniformly beneficial. In line with recent concerns about unintended emotional effects in climate communication, disaster imagery can evoke not only harm-oriented emotions[24] but also resistance-oriented reactions. To the best of our knowledge, this research is among the first to provide empirical evidence that climate disaster imagery, particularly when AI-generated, can elicit psychological reactance and reduce perceived trustworthiness of the message sender[7,31,44].

An intriguing question is whether resistance-oriented responses to AI-generated climate disaster imagery also extend beyond mass-mediated advocacy campaigns. In more participatory contexts, audiences are often explicitly invited to use generative AI to visualize familiar places affected by climate disasters[12,13]. In such co-creative settings, AI imagery functions less as a top-down persuasive device and more as a bottom-up tool for personal engagement, exploration, and sensemaking. Responses in these contexts have been associated with reflection, sadness about potential loss, and curiosity, rather than with reactance or decreased trustworthiness[12]. Tentatively, this suggests that resistance-oriented responses are most likely when AI-generated imagery is deployed by others in a top-down manner and perceived as a deliberate attempt to influence behavior. In contrast, when individuals use AI to create their own visualizations, the technology may be experienced as empowering rather than manipulative.

With regard to persuasive outcomes, our findings show that highly realistic AI-generated climate disaster images fail to yield a persuasive advantage. Notably, when their AI origin is suspected but not transparently disclosed, such images even reduce individuals' willingness to make personal sacrifices for climate action. This reaction may resemble responses to undisclosed sponsored or manipulated media content that is later recognized as such, which can trigger feelings of deception and reduce trust. Scholars have therefore argued for consistently labeling AI-generated imagery[11].

Another interesting observation concerns participants' perceptions of image origin. Although the highly realistic images used in this research retained stylized features characteristic of AI-generated imagery at the time of data collection, a substantial number of participants nonetheless perceived them as real photographs ($N = 148$ in Study 3). One explanation is that visual imagery continues to convey a strong sense of facticity, such that even in the era of AI imagery, seeing often remains believing. Alternatively, it may also reflect increasingly blurred boundaries between edited photographs and AI-generated imagery, as both can exhibit similar stylized features and may therefore be difficult to distinguish[16]. To avoid potential backlash effects when authentic photographs are mistakenly perceived as AI-generated, it may therefore be worthwhile to consider not only labeling AI-generated images, but also explicitly labeling real photographs (e.g., "verified real image"). Future research could further examine the effects of AI versus real-image labels on audience perceptions and responses to climate disaster advocacy.

The present research carries important theoretical implications and offers novel insights into the role of visual imagery in climate change

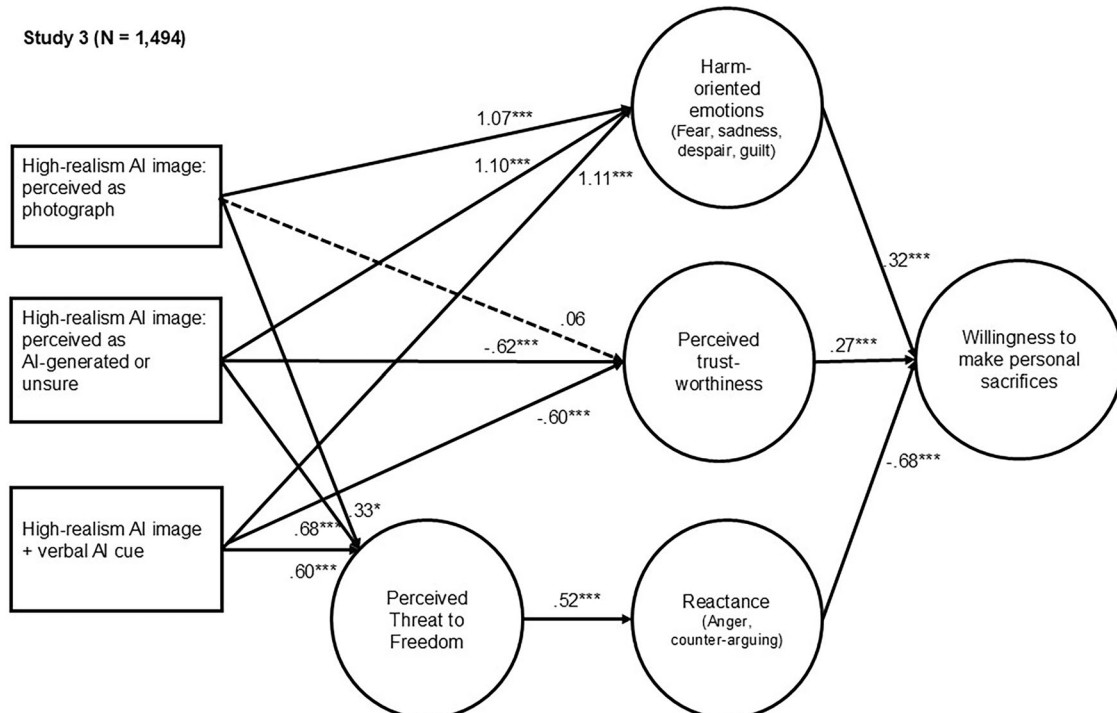

**Fig. 4 | Study 3: Structural model showing how visual realism and perceived origin in AI-generated climate disaster imagery influence individuals' willingness to make personal sacrifices for climate action.** Path coefficients are unstandardized weights. Age, gender, and political orientation were included as control variables. Dashed lines indicate nonsignificant paths. *$p \le 0.05$. **$p \le 0.01$. ***$p \le 0.001$. Model fit Study 3: $\chi^2$(362, $N = 1494$) = 2554.624, $p < 0.001$; RMSEA = 0.064 (90% confidence interval [CI] = 0.061–0.066); CFI = 0.934; and TLI = 0.920.

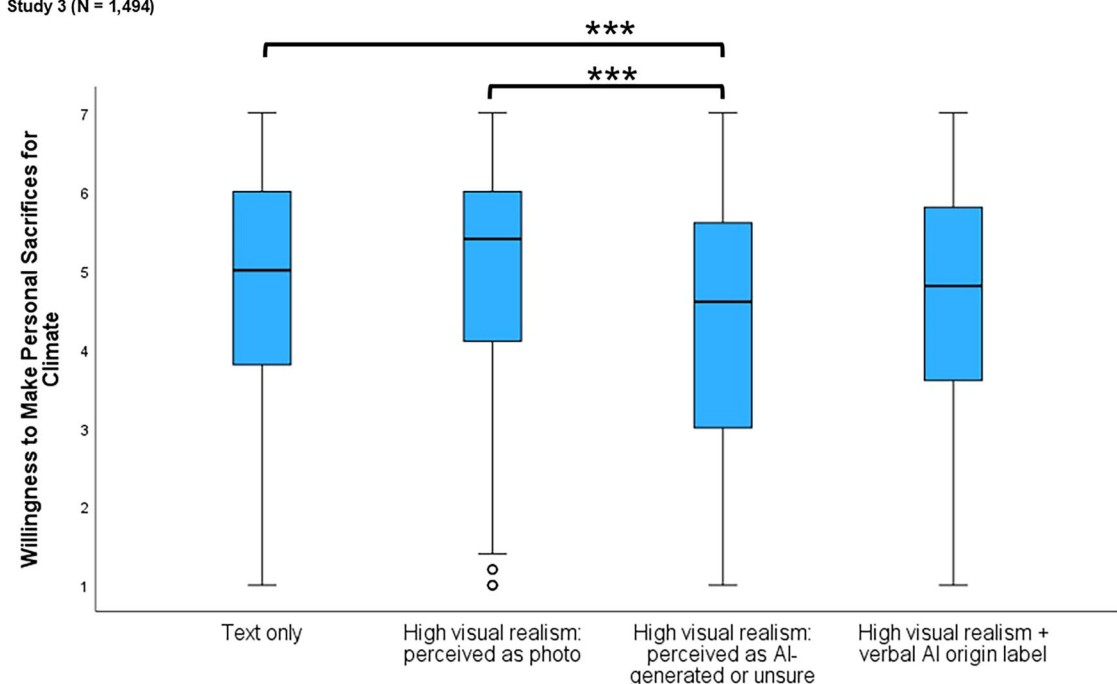

**Fig. 5 | Study 3: Box plots showing the effects of AI-generated climate disaster imagery on willingness to make personal sacrifices for climate action.** Boxes represent the interquartile range (25th–75th percentiles), center lines indicate medians, whiskers indicate the range of non-outlier values, and points represent outliers. *$p \le 0.05$. **$p \le 0.01$. ***$p \le 0.001$.

advocacy. Visuals have traditionally been treated as ornamental or illustrative campaign elements with limited capacity to trigger counter-persuasive effects on their own. Our results challenge these assumptions and contribute to a small but growing body of research documenting the potential of visual imagery to induce unintended effects and even undermine persuasive efforts in climate and social communication[7,35,40]. Importantly, our work advances this literature by identifying features of AI-generated disaster imagery that are especially likely to elicit such undesired effects and by clarifying the psychological mechanisms through which they arise in climate change advocacy.

Our findings also have clear practical implications for climate advocacy. Most importantly, they challenge the widely held assumption that highly realistic AI-generated images are inherently persuasive. While this assumption has been supported for AI-generated images depicting climate solutions—such as green urban futures[16]—it does not appear to hold for visualizations of climate disasters. At best, when mistaken for photographs, such images do not meaningfully increase support for climate action. At worst, when they are suspected to be AI-generated but not transparently disclosed, they even reduce individuals' willingness to support climate action. Climate advocates, therefore, seem to have relatively little to gain from using highly realistic AI-generated disaster imagery. In contrast, AI-generated climate disaster images with lower visual realism did not perform substantially better overall; however, their potential to elicit resistance was slightly less pronounced and less consistent. This tentatively suggests that, if AI-generated disaster images are used at all, schematic, illustrative visualizations may be preferable.

A second implication concerns the environmental impact of generative visual AI. Although the financial costs for individual users may be relatively low compared to traditional production methods or licensed image databases, these systems entail substantial environmental costs. Their use relies on resource-intensive computation, requiring considerable energy as well as water resources[45,46]. For instance, generating a single image can, in some cases, consume as much energy as charging a smartphone to around 50%[47]. Considering these environmental costs alongside the limited effectiveness of AI-generated disaster images in promoting climate action, campaign designers may therefore want to reconsider the use of generative visual AI to depict the dire consequences of climate change.

Although AI-generated images of climate disasters appear limited in their ability to motivate climate action, they may still serve to raise awareness of climate-related risks (i.e., perceptions of potential damage or harm caused by climate change)[48]. Our research does not support this assumption. Perceptions of climate risks were consistently high across all conditions in all experiments, suggesting that most participants were already aware of the harms of climate change (see additional analyses in the Appendix, Table A.13). Taken together, our findings raise serious concerns about the use of highly realistic AI-generated climate disaster images to promote climate action. Rather than increasing support, such images can undermine public willingness to make personal sacrifices for climate action. Identifying effective visual strategies is critical, but so is understanding what may backfire. This study takes an important step in that direction.

This research is subject to several limitations. First, all experiments relied on self-reported willingness to support climate action as the persuasive outcome variable. While intentions are often used as a proxy for actual behavior, future research should complement these findings with behavioral data, for instance, on consumption choices or voting patterns. Second, the effects of repeated or long-term exposure to AI-generated visuals remain unclear. As such imagery becomes more common, public responses may evolve due to habituation or shifting societal norms. Future studies could investigate how these dynamics unfold over time. Third, this study focused on U.S.-based participants. Further research is needed to explore whether these findings generalize across national and cultural contexts. Cultural differences in political systems, climate policy attitudes, and perceptions of AI may all shape how visual messages are interpreted.

Fourth, our studies focused on mass-mediated campaigns using fictional urban scenes, suggesting that reactance to AI-generated climate disaster imagery is robust even after controlling for participants' prior experience with climate disasters. An open question is whether reactance responses might vary when imagery depicts specific, personally meaningful locations. On the one hand, such depictions may reduce psychological distance and render climate consequences more tangible, personal, and immediate[12,25,44]. On the other hand, when used in top-down mass-mediated campaigns, imagery that "hits closer to home" may intensify perceptions of manipulative intent and autonomy threat by making persuasive pressure more salient. We therefore encourage future research to examine potential interaction effects between psychological distance and reactance.

## Methods
### Study 1 (Visual realism)
We recruited 535 U.S.-based, English-speaking adults (18+) from a market research panel (Prolific). After screening for data quality issues, one participant was excluded for survey rushing, resulting in a final sample of $N = 534$. Study 1 utilized a one-factor (call to action vs. call to action + low-realism AI image vs. call to action + high-realism AI image) between-subjects design. Message exposure was randomized at two levels. First, participants were randomly assigned to either the control condition, the low-realism AI-image condition, or the high-realism AI-image condition, and were then randomly assigned to one of three stimuli within each condition.

All visuals depicted catastrophic flooding. The high-realism images were generated with Midjourney and designed to resemble major cities (e.g., New York, Miami, Los Angeles). Midjourney was employed as, at the time of the study, it was a widely used image generation tool known for producing high-quality, realistic images[49]. Access to Midjourney was obtained via a paid standard subscription ($30 per month). Low-realism AI-generated images were created using OpenAI's ChatGPT (GPT-4o) via the standard interface. These images were generated by prompting the model to produce simplified, schematic representations of the same flood scenarios, thereby holding informational content constant while systematically varying visual realism. ChatGPT was used to generate low-realism images because it more reliably produced schematic, abstract outputs, whereas Midjourney tended to generate highly realistic images even when prompted otherwise. Access to ChatGPT was provided through a paid Plus subscription ($20 per month). A detailed overview of the stimulus generation and prompts used in the reported studies is provided in Appendix (Table A.2).

A pretest on Prolific ($N = 35$) confirmed that the low- and high-realism images were perceived as equally colorful, while realism ratings differed significantly (see Appendix, Table A.4). Following consent, participants were directed to a survey that captured sociodemographic information (e.g., age, gender, education, political orientation, and prior disaster experience, see Appendix, Table A.1). They were then randomly assigned to one of nine stimuli (see Appendix, Fig. A.1). After viewing one of nine messages, participants were asked about their reactions (i.e., perceived threat to freedom, emotions and reactance, and willingness to make personal sacrifices for climate action). All variables were measured using 7-point Likert scales (see Appendix, Table A.5). Items assessing each construct were averaged to form composite scores. Data analyses for the structural equation models were conducted in AMOS (version 31), and analyses of (co)variance were performed in SPSS (version 31).

### Study 2 (Replication visual realism)
We invited 553 U.S.-based, English-speaking adults (18+) from a market research panel (Prolific) to participate in the study. After screening for data quality issues, one participant was excluded for survey rushing, resulting in a final sample of $N = 552$. Study 2 employed the same one-factorial, between-subjects design as Study 1 (call to action vs. call to action + low-realism AI image vs. call to action + high-realism AI image) between-subjects design. Message exposure was again randomized at two levels.

The high-realism images were again created using text-to-image generation in Midjourney (Appendix, Table A.3). We visualized locations in

U.S. states that are particularly vulnerable to specific climate threats: wildfires (California), hurricanes (Louisiana), and coastal flooding (Florida). The low-realism AI-generated images were again created by prompting ChatGPT (GPT-4o) to produce simplified, symbolic versions of the same flood scenarios, keeping informational content constant.

A pretest conducted on Prolific ($N = 35$) confirmed that the high- and low-realism images were perceived as equally colorful, while differing significantly in perceived visual realism (see Appendix, Table A.4). Following informed consent, participants completed a survey that was identical in structure and measures to Study 1, but differed in the specific stimulus set tested (see Appendix, Fig. A.2).

## Study 3 (Perceived origin)

We recruited 1533 U.S.-based participants on Prolific and used a one-factorial between-subjects design. Participants were randomly assigned to one of three conditions: (1) a text-only message with a call to action, (2) the same message paired with a high-realism AI-generated image of a climate disaster (flood, wildfire, or hurricane), or (3) the same message and image with an added label stating the image was "created by artificial intelligence". Within each condition, participants were randomly assigned to one of three message versions.

We used the same images as in the high-realism condition in Study 2. Participants in the image conditions (2 and 3) were also asked whether they believed the image was AI-generated, a photograph, or if they were unsure. In condition 3, this question served as a manipulation check; participants who failed to identify the image as AI-generated were excluded from subsequent analyses. After excluding these participants ($n = 34$), we further excluded four participants for survey rushing and one for straightlining behavior (i.e., near-zero standard deviation across responses), resulting in a final sample of $N = 1494$. Besides perceived trustworthiness, all measures were the same as in the previous studies.

## Ethics statement

Our research was approved and exempt from formal review by the Ethics Committee of the University of St.Gallen, and was conducted in full compliance with ethical research standards (Exemption Letter, St.Gallen, 13 December 2024).

## Reporting summary

Further information on research design is available in the Nature Research Reporting Summary linked to this article.

## Data availability

The datasets supporting the findings of this research are available at https://doi.org/10.5281/zenodo.20049073.

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

## Acknowledgements
This work was supported by the Swiss National Science Foundation (SNSF) under the grant number 214958.

## Author contribution
Fabienne Bünzli: Conceptualization, Formal analysis, Methodology, Project administration, Writing—original draft, Writing—review & editing. Martin J. Eppler: Conceptualization, Validation, Writing—review & editing.

## Competing interests
The authors declare no competing interests.
