## [Transparent Peer Review File · Communications Sustainability]

Realistic AI-Generated Climate Disaster Images Decrease Support for Climate Action When Artificial Origin is Suspected

Corresponding Author: Dr Fabienne Bünzli

Version 0:

Decision Letter:

Dear Dr Bünzli,

Your manuscript titled "When AI Imagery Backfires: Photorealistic Climate Disaster Images Reduce Support for Climate Action" has now been seen by 2 reviewers, and we include their comments at the end of this message. They find your work of interest, but some important points are raised. We are interested in the possibility of publishing your study in *Communications Sustainability*, but would like to consider your responses to these concerns and assess a revised manuscript before we make a final decision on publication.

For publication in *Communications Sustainability* to be appropriate, your study must:

- * provide convincing evidence about the possible effect of AI-labeled imagery on psychological reactance and its role for support for climate action; to that effect, your manuscript must convincingly support its study design (labeling, consistency of stimuli);
- * transparently report methods and analyses to enable reproducibility—our statistical guidelines can be found at <https://www.nature.com/commssustain/submit/submission-guidelines#statistical-guidelines>; and
- * fully acknowledge limitations.

We therefore invite you to revise and resubmit your manuscript, along with a point-by-point response that takes into account the points raised. Please highlight all changes in the manuscript text file.

Please note that, in the absence of substantial revisions, we are unlikely to send your manuscript back to review.

Please submit your point-by-point responses as a separate file, distinct from your cover letter where you can add responses to the Editors' comments that you do not want to be made available to the reviewers. Word files are preferred. We recommend that any figures, tables or graphs that are included in the response to reviewers are also included in the main article or Supplementary Information.

Please use the following link to submit your revised manuscript, point-by-point response to the reviewers' comments (which should be in a separate document to any cover letter), a tracked-changes version of the manuscript (as a PDF file) and the completed checklist:

Link Redacted

We hope to receive your revised paper within six weeks; please let us know if you aren't able to submit it within this time so

that we can discuss how best to proceed. If we don't hear from you, and the revision process takes significantly longer, we may close your file. In this event, we will still be happy to reconsider your paper at a later date, as long as nothing similar has been accepted for publication at Communications Sustainability or published elsewhere in the meantime.

Please do not hesitate to contact us if you have any questions or would like to discuss these revisions further. We look forward to seeing the revised manuscript and thank you for the opportunity to review your work.

Best regards,

Madeline Judge, PhD
Editorial Board Member
Communications Sustainability
orcid.org/0000-0001-5215-9353

Yann Benetreau, PhD
Deputy Editor
Communications Sustainability
Nature Portfolio
New York City Office

EDITORIAL POLICIES AND FORMATTING

- Behavioural and social science
- Ecological, evolutionary & environmental sciences
- Life sciences

Furthermore, please align your manuscript with our format requirements, which are summarized on the following checklist:

<https://www.nature.com/documents/commsj-phys-style-formatting-checklist-article.pdf> Communications Sustainability formatting checklist

and also in our style and formatting guide <https://www.nature.com/documents/commsj-phys-style-formatting-guide-accept.pdf> Communications Sustainability formatting guide .

*** DATA: Communications Sustainability endorses the principles of the Enabling FAIR data project (<http://www.copdess.org/enabling-fair-data-project/>). We ask authors to make the data that support their conclusions available in permanent, publicly accessible data repositories. (Please contact the editor if you are unable to make your data available).

All Communications Sustainability manuscripts must include a section titled "Data Availability" at the end of the Methods section or main text (if no Methods). More information on this policy, is available at <http://www.nature.com/authors/policies/data/data-availability-statements-data-citations.pdf>.

If a community resource is unavailable, data can be submitted to generalist repositories such as <https://figshare.com/> or <http://datadryad.org/> Dryad Digital Repository. Please provide a unique identifier for the data (for example a DOI or a permanent URL) in the data availability statement, if possible. If the repository does not provide identifiers, we encourage authors to supply the search terms that will return the data. For data that have been obtained from publicly available sources, please provide a URL and the specific data product name in the data availability statement. Data with a DOI should be further cited in the methods reference section.

Please refer to our data policies at <http://www.nature.com/authors/policies/availability.html>

REVIEWER COMMENTS:

Reviewer #1 (Remarks to the Author):

This paper addresses an important and timely question about AI-generated climate imagery. I think the message of this paper is very noteworthy and very important. However, I have significant concerns with the study designs and whether the current evidence supports authors' claim at this time. Thus, it is difficult for me to recommend publication at this point. I hope my comments below are useful to the authors.

Major Concerns:

1) Stimuli consistency: The "photorealistic" images in Study 1 (Figure 1) appear quite stylized rather than truly photorealistic, undermining the visual realism manipulation (of course these images are more realistic than abstract images, but the participants' lower willingness on these images might simply be because these images didn't look real and/or looked highly edit).

More problematically, Study 2 uses entirely different images (please correct me if I am wrong though). I am not sure why this was the case and without replication using consistent stimuli, it's difficult to assess whether the effects are robust or stimulus-specific.

2) Weak statistical evidence: Multiple key findings barely reach significance (Study 1: $p=.040$, $.044$; Study 2: $p=.076$), with very small effect sizes ($\eta^2p = .012$, $.017$). The lack of pre-registration raises questions about analytic flexibility and multiple comparisons. Furthermore, one key finding from Study 2 is a bit inconsistent with the message of the paper— the images only backfire when suspected to be AI-generated, not when labeled as such, which seems rather puzzling ($p=.076$, not significant). Given the importance of these effects for the paper's argument, pre-registration and/or replication with consistent stimuli would be valuable.

3) Measurement details: The appendix lists multiple items under "willingness to make personal sacrifices," but I couldn't find clarification in the methods about whether these were averaged or analyzed separately. A bit more detail here would be helpful.

Summary: While the research question has merit and the large-scale experiments are very noteworthy, unfortunately, the inconsistent findings, marginal statistics, and lack of replication make the evidence insufficient for publication. I would encourage the authors to conduct pre-registered replications with consistent stimuli and clearer theoretical predictions before resubmission.

Reviewer #2 (Remarks to the Author):

Summary: This paper presents two experimental studies ($N = 2,028$) investigating how AI-generated photorealistic images of climate disasters affect public support for climate action. Results show that such images can decrease willingness to act, primarily due to perceptions of manipulation and lower trust when viewers suspect or know the image is AI-generated.

Strengths:

- Clear design and analyses: Two well-executed experiments using realistic image manipulations and appropriate statistical testing (ANOVA, SEM).
- Timely topic: Addresses the growing use of AI-generated visuals in environmental communication.
- Theoretical contribution: Integrates visual persuasion and reactance theory to explain counter-persuasive effects.
- Practical relevance: Offers concrete caution for communicators and NGOs experimenting with AI visuals.

Weaknesses / Areas for Improvement

Contextual variables omitted:

The study does not consider place attachment or psychological distance, both known to shape responses to climate imagery (e.g., Van Lange & Huckelba, 2021; Singh et al., 2017; Brügger, 2020).

Limited ecological validity:

Participants saw generic, distant urban scenes. Reactions may differ when the setting is personally meaningful or local. This is also closely related to place attachment and psychological distance

Terminology:

The paper refers to an “abstract” condition, but the images appear to be stylized rather than abstract. “Stylized” would better capture the difference in visual treatment (artistic vs. photorealistic) and avoid confusion about the nature of the manipulation.

Narrow interpretation of results:

The paper frames realism as inherently risky but does not explore when or how AI-generated visuals might enhance engagement—e.g., through participatory or co-creative approaches.

Missing connection to emerging work:

The authors should cite and discuss:

Castillo, L., Rafner, J. F., Nockur, L. E., & Buttlar, B. (2024). Seeing the World Submerged: Exploring the Potential of AI Image Generation for Climate Change Communication (preprint).

This study explicitly examines AI imagery, place attachment, and emotional engagement—offering a relevant counterpoint.

Minor comment (terminology)

The paper describes a condition using “abstract” AI-generated images. However, these visuals depict recognizable urban environments and are therefore stylized, not abstract.

Abstract: Images that do not represent recognizable objects or scenes from the real world, focusing on shapes, colors, and composition.

Stylized: Images that do represent real-world subjects but through an artistic style that simplifies, exaggerates, or distorts features.

Given that the study’s “low realism” condition retains identifiable city elements, “stylized” is the more accurate term and should replace “abstract” throughout the manuscript.

Relevant articles:

Metag, J., Schäfer, M. S., Fuchslin, T., Barsuhn, T., & Kleinen-von Königslöw, K. (2016). Perceptions of climate change imagery. *Science Communication*, 38(2), 197–227. <https://doi.org/10.1177/1075547016635181>;

Wong-Parodi, G., & Feygina, I. (2021). Engaging people on climate change: The role of emotional responses. *Environmental Communication*, 15(5), 571–593. <https://doi.org/10.1080/17524032.2020.1871051>;

Wang, S., Corner, A., Chapman, D., & Markowitz, E. (2018). Public engagement with climate imagery in a changing digital landscape. *WIREs Climate Change*, 9(2). <https://doi.org/10.1002/wcc.509> for psychological distance place refer to this: Van Lange, P. A. M., & Huckelba, A. L. (2021).

Psychological distance: How to make climate change less abstract and closer to the self. *Current Opinion in Psychology*, 42, 49–53. <https://doi.org/10.1016/j.copsyc.2021.03.011> and this Singh, A. S.,

Zwickle, A., Bruskotter, J. T., & Wilson, R. (2017). The perceived psychological distance of climate change impacts and its influence on support for adaptation policy. *Environmental Science & Policy*, 73, 93–99. <https://doi.org/10.1016/j.envsci.2017.04.011>

Metag, J., Schäfer, M. S., Fuchslin, T., Barsuhn, T., & Kleinen-von Königslöw, K. (2016). Perceptions of climate change imagery. *Science Communication*, 38(2), 197–227. <https://doi.org/10.1177/1075547016635181> and this one on concrete vs. abstract images.

Duan, R., Takahashi, B., & Zwickle, A. (2021). How effective are concrete and abstract climate change images? The moderating role of construal level in climate change visual communication. *Science Communication*, 43(3), 358–387. <https://doi.org/10.1177/10755470211008192> and this one on psychological distance:

Brügger, A. (2020). Understanding the psychological distance of climate change: The limitations of construal level theory and suggestions for alternative theoretical perspectives. *Global Environmental Change*, 60, 102023. <https://doi.org/10.1016/j.gloenvcha.2019.102023>. this one on local framing and place attachment.

** Visit Nature Portfolio's author and reviewers' website at www.nature.com/authors for information about policies, services and author benefits**

Communications Sustainability is committed to improving transparency in authorship. As part of our efforts in this direction, we are now requesting that all authors identified as 'corresponding author' create and link their Open Researcher and Contributor Identifier (ORCID) with their account on the Manuscript Tracking System prior to acceptance. ORCID helps the scientific community achieve unambiguous attribution of all scholarly contributions. You can create and link your ORCID from the home page of the Manuscript Tracking System by clicking on 'Modify my Springer Nature account' and following the instructions in the link below. Please also inform all co-authors that they can add their ORCIDs to their accounts and that they must do so prior to acceptance.

Version 1:

Decision Letter:

Dear Dr Bünzli,

Your revised manuscript titled "Unintended Effects of AI-Generated Imagery: Realistic Climate Disaster Images Reduce Support for Climate Action" has now been seen by the same 2 reviewers, and we include their comments at the end of this message. While both reviewers appreciated the improvements made in the last revision, they still have some remaining concerns to be addressed. In particular, both reviewers note the need for the interpretations to be more closely tied to the findings.

We therefore invite you to revise and resubmit your manuscript again, along with a point-by-point response that takes into account the points raised. Please highlight all changes in the manuscript text file.

Please submit your point-by-point responses as a separate file, distinct from your cover letter where you can add responses to the Editors' comments that you do not want to be made available to the reviewers. Word files are preferred. We recommend that any figures, tables or graphs that are included in the response to reviewers are also included in the main article or Supplementary Information.

Please use the following link to submit your revised manuscript, point-by-point response to the reviewers' comments (which should be in a separate document to any cover letter), a tracked-changes version of the manuscript (as a PDF file) and the completed checklist:

Link Redacted

We hope to receive your revised paper within six weeks; please let us know if you aren't able to submit it within this time so that we can discuss how best to proceed. If we don't hear from you, and the revision process takes significantly longer, we may close your file. In this event, we will still be happy to reconsider your paper at a later date, as long as nothing similar has been accepted for publication at Communications Sustainability or published elsewhere in the meantime.

Please do not hesitate to contact us if you have any questions or would like to discuss these revisions further. We look forward to seeing the revised manuscript and thank you for the opportunity to review your work.

Best regards,

Madeline Judge, PhD
Editorial Board Member
Communications Sustainability
orcid.org/0000-0001-5215-9353

Yann Benetreau, PhD
Deputy Editor, Communications Sustainability
Consulting Editor, Communications Earth & Environment
Nature Portfolio
NY office

EDITORIAL POLICIES AND FORMATTING

- Behavioural and social science
- Ecological, evolutionary & environmental sciences
- Life sciences

Furthermore, please align your manuscript with our format requirements, which are summarized on the following checklist: [Communications Sustainability formatting checklist](https://www.nature.com/documents/commsj-phys-style-formatting-checklist-article.pdf)

and also in our style and formatting guide [Communications Sustainability formatting guide](https://www.nature.com/documents/commsj-phys-style-formatting-guide-accept.pdf) .

*** DATA: Communications Sustainability endorses the principles of the Enabling FAIR data project (<http://www.copdess.org/enabling-fair-data-project/>). We ask authors to make the data that support their conclusions available in permanent, publicly accessible data repositories. (Please contact the editor if you are unable to make your data available).

All Communications Sustainability manuscripts must include a section titled "Data Availability" at the end of the Methods section or main text (if no Methods). More information on this policy, is available at <http://www.nature.com/authors/policies/data/data-availability-statements-data-citations.pdf>.

If a community resource is unavailable, data can be submitted to generalist repositories such as [figshare](https://figshare.com/) or [Dryad Digital Repository](http://datadryad.org/). Please provide a unique identifier for the data (for example a DOI or a permanent URL) in the data availability statement, if possible. If the repository does not provide identifiers, we encourage authors to supply the search terms that will return the data. For data that have been obtained from publicly available sources, please provide a URL and the specific data product name in the data availability statement. Data with a DOI should be further cited in the methods reference section.

REVIEWER COMMENTS:

Reviewer #1 (Remarks to the Author):

I appreciate the authors' efforts in revising the manuscript and conducting an additional replication study. I thank them for taking my concerns seriously and I can now more clearly see a path towards publication of the paper. However, I still have several substantive concerns that I think should be address before publication.

Major concerns:

1. The Paper Overstates the "Backfire" Effect

Now that the authors have conducted an additional replication study, to me, the manuscript's central claim -- that highly realistic AI disaster images reduce support for climate action -- is not consistently supported and reflects a misleading emphasis on SEM path coefficients over actual total effects. The ANCOVA results tell a different story:

Study 1: High realism vs. text-only, $p = .208$ -- this is null effect but NOT a backfire. I do note that the low realism condition moves the needle in the positive direction ($p = .044$), which is very interesting; but this is not evidence for a backfire effect.

Study 2: No significant effect of condition on willingness ($p = .156$). Again, here there is no backfire.

Study 3: A backfire emerges only when participants spontaneously suspected AI origin ($p < .001$ vs. text-only). The explicitly labeled condition does not reach significance ($p = .076$ — see the minor concern section).

Thus, here, the accurate summary is: High-realism AI disaster images consistently fail to increase willingness, but do not reliably reduce it. In Study 1, the low realism condition moves the willingness needle, which is an interesting result. Backfire is a conditional effect specific to Study 3, and only among participants who detected AI origin unprompted. The current presentation of the paper obscures this by foregrounding significant SEM indirect paths as evidence of backfire, while treating the null ANCOVA results as secondary. Significant indirect paths are not the same as a significant total effect. The title, abstract, and discussion should be revised to reflect that the primary finding is a failure to persuade rather than active backfire.

2. The authors should clarify that images are not photorealistic by contemporary standards

I appreciate the authors clarification of the "high realism" images. However, it should be noted that the "high realism" images used in the Experiments remain visibly stylized — dramatic lighting, saturated color grading, and an aesthetic characteristic of AI art generation circa 2022-2023. These are not photorealistic in the sense that contemporary audiences would mistake for photographs. Modern AI systems are now capable of producing genuinely photo-indistinguishable imagery. The manuscript warns against "highly realistic" AI imagery, but the stimuli used here would likely be immediately detectable as AI-generated today.

All applied recommendations must be qualified accordingly: these findings apply to stylized-yet-detailed AI imagery that is detectable as AI-generated, and may not generalize to truly photorealistic outputs. I think the introduction and discussions should highlight this important caveat so that the paper findings are more readily useful for practitioners.

3. Title and Abstract Must Reflect Actual Scope and Findings

The paper is specifically about AI-generated climate *disaster* imagery, and the backfire effect is conditional on perceived AI origin. Neither the title nor the abstract reflects this adequately. The paper should also clarify throughout the paper that the findings are specifically about AI-generated climate disaster images only (and not about AI-generated imagery for all kinds of climate communication) and the abstract should also clarify that backfire is conditional, not general.

Other Concerns:

1. Causal Language in the Mediation Claims

The authors describe their central finding as: "highly realistic AI-generated disaster imagery intensifies emotional responses while simultaneously increasing perceived freedom threat and psychological reactance, which in turn reduce willingness to make personal sacrifices for climate action." This implies a causal chain the data cannot support. Random assignment permits causal inference for the manipulation's effects, but the mediation pathways in the SEM are correlational. The authors should replace causal language ("which in turn reduce") with associative language ("which were in turn correlated with lower willingness") throughout.

2. "Marginally Significant" Language

The authors describe the AI origin label condition vs. text-only in Study 3 as producing a "marginally significant negative effect" ($p = .076$). In my opinion, this framing should be removed. $p = .076$ is not significant, and "marginal significance" is not an accepted statistical concept (if the authors want to establish this point, then they should replicate this). This should be reported as a null result.

Summary

The research question is important and the three-study structure is a strength. However, the paper currently overstates its findings through causal language, selective emphasis on SEM paths over null total effects, and insufficiently qualified claims about photorealism and climate communication in general. These issues affect the paper's central claims and must be addressed before publication.

Reviewer #2 (Remarks to the Author):

The revision substantially improves the manuscript. In particular, the added replication study, clarified terminology around visual realism, expanded theoretical framing, and fuller reporting of measures and SEM structure address several important reviewer concerns. However, some issues remain only partially resolved. The evidence for downstream persuasive "backfire" effects is still modest and somewhat uneven across studies, the concern about lack of preregistration is not fully addressed, ecological validity is acknowledged more than empirically resolved, and the methods for generating the AI

images remain insufficiently documented for full reproducibility. I therefore view the manuscript as significantly strengthened, but still in need of greater methodological transparency and slightly more cautious interpretation of the findings.

In particular, while the revision improves the conceptual framing, the manuscript still lacks sufficient transparency about stimulus generation. Because the core manipulation depends on AI-generated imagery, the image-production pipeline is part of the method and should be documented in much greater detail. The authors should report the exact prompts used to generate each stimulus set, preferably in an appendix or repository, along with any iterative prompt refinement, selection criteria, relevant generation settings, and whether any post-processing occurred. The manuscript should also explain why Midjourney was used for some images and ChatGPT for others, and specify exactly which ChatGPT image model/version was used. Relatedly, if access depended on a paid subscription, this should be disclosed as part of the methods, since it affects reproducibility.

Finally, the framing of AI image generation as low-cost or free should be qualified. Even if it is financially inexpensive relative to traditional production methods, GenAI image generation is not costless. It carries environmental impacts associated with inference-time computation, including energy and water use. A brief acknowledgment of this limitation would improve the paper's balance and accuracy. This is particularly relevant as it is about climate change...

See the following references:

Utz, V. and DiPaola, S. 2023. Climate Implications of Diffusion-based Generative Visual AI Systems and their Mass Adoption. In Proceedings of the 14th International Conference on Computational Creativity (ICCC '23), 264–272.

Emma Strubell, Ananya Ganesh, and Andrew McCallum. 2019. Energy and Policy Considerations for Deep Learning in NLP. In Proceedings of the 57th Annual Meeting of the Association for Computational Linguistics (ACL 2019), 3645–3650. <https://doi.org/10.18653/v1/P19-1355>

International Telecommunication Union (ITU). 2025. Measuring what matters: How to assess AI's environmental impact (S-GEN-GDA.001-2025). Retrieved January 23, 2026 from <https://www.itu.int/hub/publication/s-gen-gda-001-2025/>

Rafner J., & Gibat C., Every Prompt Has a Price A Toolbox for Tracking the Environmental Cost of GenAI Use. CHI 2026

** Visit Nature Portfolio's author and reviewers' website at <http://www.nature.com/authors> for information about policies, services and author benefits**

Communications Sustainability is committed to improving transparency in authorship. As part of our efforts in this direction, we are now requesting that all authors identified as 'corresponding author' create and link their Open Researcher and Contributor Identifier (ORCID) with their account on the Manuscript Tracking System prior to acceptance. ORCID helps the scientific community achieve unambiguous attribution of all scholarly contributions. You can create and link your ORCID from the home page of the Manuscript Tracking System by clicking on 'Modify my Springer Nature account' and following the instructions in the link below. Please also inform all co-authors that they can add their ORCIDs to their accounts and that they must do so prior to acceptance.

Version 2:

Decision Letter:

Dear Dr Bünzli,

Thank you for submitting a revised version of your manuscript titled "Realistic AI-Generated Climate Disaster Images Decrease Support for Climate Action When Artificial Origin is Suspected." We are delighted to say that we are happy, in principle, to publish a suitably revised version in Communications Sustainability.

We therefore invite you to revise your paper one last time to address remaining editorial concerns. At the same time we ask

that you edit your manuscript to comply with our format requirements and to maximise the accessibility and therefore the impact of your work.

EDITORIAL REQUESTS:

****Please take care to match our formatting and policy requirements. We will check revised manuscript and return manuscripts that do not comply. Such requests will lead to delays. ****

SUBMISSION INFORMATION:

OPEN ACCESS:

Communications Sustainability is a fully open access journal. Articles are made freely accessible on publication. For further information about article processing charges, open access funding, and advice and support from Nature Portfolio, please visit <https://www.nature.com/commssustain/open-access>

Link Redacted

Best regards,

Madeline Judge, PhD
Editorial Board Member
Communications Sustainability
orcid.org/0000-0001-5215-9353

Yann Benetreau, PhD
Consulting Editor, Communications Earth & Environment
Deputy Editor, Communications Sustainability
Nature Portfolio
NY office

** Visit Nature Portfolio's author and reviewers' website at <http://www.nature.com/authors> for information about policies, services and author benefits**

General Remarks

We wish to thank the editors, Dr. Judge and Dr. Benetreau, as well as the two reviewers for their thoughtful and constructive comments on the manuscript. We sincerely appreciate the time and effort dedicated to the review process, the meticulous reading of our work, and the nuanced feedback provided.

The reviews offered valuable guidance on how to strengthen the manuscript on the persuasive effects of AI-generated climate disaster imagery. We are also grateful for the literature recommendations, which we have incorporated into the revised manuscript.

Guided by the editorial and reviewer feedback, we have substantially revised the manuscript. In the following, we respond point-by-point to the comments from the editors and the two reviewers. All substantial changes in the manuscript are highlighted in yellow.

Editors	Our Response
We are interested in the possibility of publishing your study in Communications Sustainability, but would like to consider your responses to these concerns and assess a revised manuscript before we make a final decision on publication. For publication in Communications Sustainability to be appropriate, your study must:  * provide convincing evidence about the possible effect of AI-labeled imagery on psychological reactance and its role for support for climate action; to that effect, your manuscript must convincingly support its study design (labeling, consistency of stimuli); 	We are delighted that you see merit in our work, and we sincerely thank you for the valuable guidance on how to strengthen the paper. We have taken your concerns to heart and have substantially revised the manuscript to address them. To further strengthen the evidence for the reactance-inducing potential of AI-generated disaster imagery, we conducted a replication of Study 1 (now presented as Study 2 in the revised manuscript) with N = 553 participants). The findings provide further support that AI-generated disaster images—especially those high in visual realism—elicit psychological reactance, which subsequently reduces support for climate action. Across all three studies (total N = 2,580), reactance is consistently negatively associated with individuals’ willingness to make personal sacrifices for climate action. Moreover, we expanded the theoretical section to more clearly explain how AI-generated climate disaster images heighten perceptions of persuasive pressure and freedom threat, thereby triggering reactance. These revisions strengthen the theoretical underpinning of the findings reported in the manuscript. Following your guidance, we also refined the labeling and aligned the terminology throughout the manuscript. In particular, we aligned our wording more closely with the established terminology of psychological reactance research. For instance, what we previously referred to in more lay terms as “perceived manipulation” is now consistently labeled as perceived threat to freedom, which is the

	standard expression used in reactance literature (Brehm & Brehm, 1981; Dillard & Shen, 2005; Jenkins & Dragojevic, 2011). Importantly, this reflects a terminological clarification and not a change in measurement: the underlying items and operationalization remained the same (see Appendix Table A.3, based on Dillard & Shen’s seminal scale). Based on feedback from Reviewers 1 and 2, we also revised the labels for our visual realism manipulation to improve conceptual clarity. Rather than referring to the stimuli as “photographic” versus “abstract,” we now describe them more precisely as images high versus low in visual realism. This terminology better captures the intended continuum of lifelikeness and photographic fidelity, without implying that the low-realism stimuli are purely abstract or devoid of recognizable content. We also describe in more detail the cues that distinguish images that are high versus low in visual realism. In addition, we have expanded the manuscript’s definition of visual realism and clarified how it differs from concepts such as stylization. Regarding stimulus consistency, we followed your suggestion and that of Reviewer 1 and replicated Study 1 using the stimulus set employed in the subsequent experiment (now Study 3). Based on this stimulus set, we were able to replicate the main findings. Moreover, the effect sizes were comparable in magnitude, further supporting the robustness of the results.
* transparently report methods and analyses to enable reproducibility—our statistical guidelines can be found at https://www.nature.com/commssustainability/submit/submission-guidelines#statistical-guidelines;	We have carefully reviewed the guidelines and revised the manuscript accordingly. For instance, we enhanced the transparency of our structural equation models by explicitly reporting exogenous and endogenous variables and clarifying the focal comparisons of interest. We also streamlined the analytic approach by applying a consistent contrast-testing strategy across studies, systematically comparing experimental conditions against the control condition. In addition, we clarified and presented the full set of hypotheses guiding our research. For all analyses, we now report exact p-values.

* fully acknowledge limitations.	Based on Reviewer 2’s comments, we expanded the limitations section to discuss the relatively generic nature of our AI-generated disaster images and the possibility that psychological distance may interact with psychological reactance in shaping responses to such imagery (see p. 18). We also sharpened the scope of the manuscript by distinguishing between two use cases of AI-generated disaster imagery. The first involves participatory, co-creative initiatives in which audiences use AI to generate individualized visualizations of climate disaster scenarios affecting local neighborhoods or personally meaningful places (see p. 3). The second concerns mass-mediated advocacy campaigns, in which AI-generated disaster images are presented to broad audiences in a top-down manner to highlight more generalized consequences of climate change. We provide real-world examples of both contexts and clarify that the manuscript focuses on the latter—AI-generated imagery in mass-mediated climate advocacy campaigns.
Reviewer 1	Our Response
This paper addresses an important and timely question about AI-generated climate imagery. I think the message of this paper is very noteworthy and very important. However, I have significant concerns with the study designs and whether the current evidence supports authors' claim at this time. Thus, it is difficult for me to recommend publication at this point. I hope my comments below are useful to the authors.	We are very grateful that you find our work of interest, and we would like to thank you for your careful reading of the manuscript and your helpful feedback. We have made a considerable effort to address your concerns, including conducting a large-scale replication study to further demonstrate the robustness of our findings and the soundness of our claims.
Major Concerns: 1) Stimuli consistency: The "photorealistic" images in Study 1 (Figure 1) appear quite stylized rather than truly photorealistic, undermining the visual realism manipulation (of course these images are more realistic than abstract	We appreciate this insightful comment, which highlighted the need to clarify our visual realism manipulation. We therefore clarified our understanding of visual realism by providing a clearer and more comprehensive definition of the concept (pp. 4 to 5). We define visual realism as the extent to which an image preserves perceptual cues of physical reality, such as naturalistic texture, depth, lighting,

images, but the participants' lower willingness on these images might simply be because these images didn't look real and/or looked highly edit).

and atmospheric detail (Blondé & Girandola, 2018; Dubey et al., 2024), and we describe in more detail the cues that distinguish images high versus low on this dimension.

We also added a clarification regarding the relationship between visual realism and stylization. While visual realism concerns lifelikeness and photographic fidelity, stylization refers to artistic transformations (e.g., filters, dramatic color grading, exaggerated contrast) that may occur at both low and high levels of realism.

Finally, to avoid overstatement and improve conceptual consistency, we adjusted the stimulus labels throughout the manuscript. Rather than referring to the images as “photographic” versus “abstract,” we now describe them more precisely as images high versus low in visual realism, which better reflects the intended continuum and improves interpretability without altering the underlying manipulation.

You noted that images low in visual realism may be less persuasive because they do not look fully real or lifelike. Indeed, this is central to our theoretical argument. Lower visual realism reduces the intensity with which disaster imagery evokes emotions such as fear, sadness, despair, guilt, and anxiety. At the same time, it also attenuates resistance-oriented responses, including anger and psychological reactance. In this sense, our findings suggest that climate disaster images differ systematically in the emotional processes they elicit: highly realistic depictions generate stronger emotional impact, but they are also more likely to activate resistance, whereas less realistic depictions produce weaker effects on both pathways.

More problematically, Study 2 uses entirely different images (please correct me if I am wrong though). I am not sure why this was the case and without replication using consistent stimuli, it's difficult to assess whether the effects are robust or stimulus-specific.

Thank you for raising this important point. You are correct that, in the original manuscript, Study 2 employed a different set of visual stimuli than Study 1. We agree that this design choice makes it harder to determine whether the observed effects are robust or specific to the particular stimuli used.

We took this comment to heart and conducted an additional large-scale replication study (N = 553), which is now presented as Study 2 in the revised manuscript. This replication was designed

	specifically to strengthen confidence in the robustness of the effects. In this new Study 2, we used the same high–visual realism disaster images that were subsequently employed in Study 3 (i.e., the perceived-origin study with N = 1,494) and created corresponding low–visual realism versions of these images. Importantly, the findings closely replicate the central pattern observed in Study 1: highly realistic AI-generated disaster imagery intensifies emotional responses while simultaneously increasing perceived freedom threat and psychological reactance, which in turn reduce willingness to make personal sacrifices for climate action. These results suggest that the key effects are not driven by idiosyncrasies of a particular image set but generalize across different disaster contexts and stimulus materials. Overall, the revised manuscript now presents three consecutive studies with a clearer and more coherent stimulus structure: Study 1 establishes the visual realism effect, Study 2 provides a direct replication with an alternative stimulus set to demonstrate robustness across materials, and Study 3 extends the findings by examining the role of perceived origin using the same high-realism images as in the previous study.
2) Weak statistical evidence: Multiple key findings barely reach significance (Study 1: $p=.040$, $.044$; Study 2: $p=.076$), with very small effect sizes ($\eta^2p = .012$, $.017$). The lack of pre-registration raises questions about analytic flexibility and multiple comparisons. Furthermore, one key finding from Study 2 is a bit inconsistent with the message of the paper— the images only backfire when suspected to be AI-generated, not when labeled as such, which seems rather puzzling ($p=.076$, not significant). Given the importance of these effects for the paper's argument, pre-registration and/or replication with consistent stimuli would be valuable.	We acknowledge that the overall effects on downstream persuasive outcomes (e.g., willingness to make personal sacrifices) are modest in magnitude. However, such effect sizes are common in experimental visual communication research, where participants are exposed to a single stimulus (Seo, 2020). Importantly, even small shifts in willingness to support climate action can be meaningful in aggregate in the context of mass-mediated advocacy. At the same time, we would like to emphasize that the primary contribution of the manuscript lies in identifying and empirically testing the psychological processes that underlie responses to AI-generated disaster imagery. Across studies, we find consistent evidence that highly realistic AI-generated disaster images elicit both desired emotional responses and as well as undesired responses, with the latter offsetting persuasive

benefits. In direct response to your call for stronger evidence and greater robustness, we conducted an additional large-scale replication study with consistent stimuli (N = 553, now Study 2 in the revised manuscript). This replication closely confirms the central pattern observed in Study 1 and demonstrates that the key effects are not stimulus-specific.

Regarding the finding that backlash effects were stronger when images were suspected to be AI-generated than when they were explicitly labeled, we agree that this pattern merits careful interpretation. As discussed in the revised manuscript (p. 16), one explanation is that suspicion without transparent disclosure may heighten perceptions of deception or strategic manipulation, thereby increasing reactance. In contrast, explicit labeling may reduce ambiguity and attenuate such responses, consistent with recent calls for transparency in AI-generated media content (O’Neill, 2025).

Finally, we substantially strengthened the theoretical framework as well as our analytic approach.

For instance, we enhanced the transparency of our structural equation models by explicitly reporting exogenous and endogenous variables and clarifying the focal comparisons of interest. We also streamlined the analytic approach by applying a consistent contrast-testing strategy across studies, systematically comparing experimental conditions against the control condition. In addition, we clarified and presented the full set of hypotheses guiding our research. Moreover, for all analyses, we now report exact p-values. We believe these revisions address concerns about interpretability and analytic flexibility and provide a coherent and robust empirical foundation for our conclusions.

Seo, K. (2020). Meta-analysis on visual persuasion. Does adding images to texts influence persuasion? *Athens Journal of Mass Media and Communications*, 6(3), 177–190.
<https://doi.org/10.30958/ajmmc.X-Y-Z>

3) Measurement details: The appendix lists multiple items under "willingness to make personal sacrifices," but I couldn't find clarification in the methods about whether these were averaged or analyzed separately. A bit more detail here would be helpful.	Thank you for this important suggestion. We have now added the requested information in the Methods section (pp. 19 to 20, Study 1), where we report that items assessing each construct, including willingness to make personal sacrifices, were averaged to form composite scores.
Summary: While the research question has merit and the large-scale experiments are very noteworthy, unfortunately, the inconsistent findings, marginal statistics, and lack of replication make the evidence insufficient for publication. I would encourage the authors to conduct pre-registered replications with consistent stimuli and clearer theoretical predictions before resubmission.	We are grateful for your guidance, which helped us improve the rigor, clarity, and overall contribution of the manuscript. We hope that the revisions have addressed your concerns to your satisfaction.
Reviewer 2	Our Response
Strengths:  -Clear design and analyses: Two well-executed experiments using realistic image manipulations and appropriate statistical testing (ANOVA, SEM). -Timely topic: Addresses the growing use of AI-generated visuals in environmental communication. -Theoretical contribution: Integrates visual persuasion and reactance theory to explain counter-persuasive effects. -Practical relevance: Offers concrete caution for communicators and NGOs experimenting with AI visuals. 	We are very grateful for your positive and encouraging feedback and for offering valuable guidance on how to strengthen the paper. We have also read the literature you recommended with great interest and incorporated it into the revised manuscript.
Weaknesses / Areas for Improvement Contextual variables omitted: The study does not consider place attachment or psychological distance, both known to shape responses to climate imagery (e.g.,	We appreciate this important point and agree that place attachment and psychological distance are relevant contextual variables that may shape responses to climate disaster imagery (e.g., Brügger, 2020). We addressed this point both empirically and conceptually. Empirically, we did so by including

Van Lange & Huckelba, 2021; Singh et al., 2017; Brügger, 2020).	an additional control variable in our statistical analyses. Specifically, in Studies 1 and 2 we assessed whether participants had been recently exposed to climate disasters (e.g., whether floods or other disasters had occurred in their local area). We used this measure as a proxy for experiential proximity to climate disasters. Importantly, controlling for this variable did not change the direction or significance of the reported effects, providing additional evidence for the robustness of our findings. At the same time, we acknowledge that this measure does not fully capture psychological distance or place attachment as conceptualized in the broader climate communication literature. We therefore expanded the limitations section (p. 18) to explicitly discuss the omission of these constructs and to highlight them as important avenues for future research. Moreover, we sharpened the scope of the manuscript by distinguishing between different use cases of AI-generated disaster imagery. Place attachment and psychological distance may be particularly consequential in participatory, co-creative initiatives, where audiences generate highly localized visualizations of climate impacts affecting personally meaningful places (see p. 3). In contrast, the present research focuses on mass-mediated advocacy campaigns, which rely on more generalized, non-site-specific depictions intended for broad audiences. We provide real-world examples of both contexts and clarify that our conclusions are most directly applicable to the latter (see pp. 3 to 4).
Limited ecological validity: Participants saw generic, distant urban scenes. Reactions may differ when the setting is personally meaningful or local. this is also closely related to place attachment and psychological distance	We acknowledge this important point regarding ecological validity. Responses to AI-generated disaster imagery may indeed differ when images depict personally meaningful or local settings, which is closely related to place attachment and psychological distance. However, our study was intentionally designed to examine AI-generated imagery in the context of mass-mediated advocacy campaigns, where visuals are typically aimed at broad audiences and often rely on more generic, non-site-specific disaster depictions.

	In the revised manuscript, we therefore sharpened the scope of our contribution by distinguishing between two key use cases of AI-generated disaster imagery. The first involves participatory, co-creative initiatives, in which audiences use AI as a tool to generate individualized visualizations of climate disaster scenarios affecting local neighborhoods or personally meaningful places (see p. 3). The second concerns mass-mediated advocacy campaigns, in which AI-generated disaster images are presented in a top-down manner to highlight more generalized consequences of climate change (see p. 3 to 4). We provide real-world examples of both contexts and clarify that the present manuscript focuses specifically on the latter, while highlighting localized and personally meaningful imagery as an important direction for future research.
Terminology: The paper refers to an “abstract” condition, but the images appear to be stylized rather than abstract. “Stylized” would better capture the difference in visual treatment (artistic vs. photorealistic) and avoid confusion about the nature of the manipulation. Minor comment (terminology) The paper describes a condition using “abstract” AI-generated images. However, these visuals depict recognizable urban environments and are therefore stylized, not abstract. Abstract: Images that do not represent recognizable objects or scenes from the real world, focusing on shapes, colors, and composition. Stylized: Images that do represent real-world subjects but through an artistic style that simplifies, exaggerates, or distorts features. Given that the study’s “low realism”	We thank you for raising this important point. Your comment, along with R1’s prompted us to revise the labels for our visual realism manipulation to improve conceptual clarity. Rather than referring to the stimuli as “photographic” versus “abstract,” we now describe them more precisely as images high versus low in visual realism. This terminology better captures the intended continuum of lifelikeness and photographic fidelity, without implying that the low-realism stimuli are purely abstract or devoid of recognizable content. In addition, we expanded the manuscript’s definition of visual realism, describe in more detail the cues that distinguish images high versus low on this dimension, and clarified how visual realism differs from concepts such as stylization.

condition retains identifiable city elements, “stylized” is the more accurate term and should replace “abstract” throughout the manuscript.	
Narrow interpretation of results: The paper frames realism as inherently risky but does not explore when or how AI-generated visuals might enhance engagement—e.g., through participatory or co-creative approaches.	We are grateful for this important point. It highlighted the need to more clearly situate our findings within the specific use case of mass-mediated advocacy campaigns and to avoid an overly narrow interpretation of visual realism as inherently risky across all contexts. In the revised manuscript, we now explicitly present participatory and co-creative applications of AI-generated disaster imagery in the Introduction section, including real-world examples (see p. 3). We also expanded the Discussion section to critically examine potential differences between mass-mediated campaign contexts and co-creative settings in which audiences actively generate imagery themselves (see pp. 15 to 16).
Missing connection to emerging work: The authors should cite and discuss: Castillo, L., Rafner, J. F., Nockur, L. E., & Buttlar, B. (2024). Seeing the World Submerged: Exploring the Potential of AI Image Generation for Climate Change Communication (preprint). This study explicitly examines AI imagery, place attachment, and emotional engagement—offering a relevant counterpoint.	Thank you very much for this helpful reference. The study is highly topical and offers an important complementary perspective by examining AI-generated imagery in relation to place attachment and emotional engagement. We have incorporated this work into the revised manuscript and discuss it as a valuable complementary perspective that helps situate our findings within the broader emerging literature (e.g., see p. 3).
Relevant articles: Metag, J., Schäfer, M. S., Füchslin, T., Barsuhn, T., & Kleinen-von Königslöw, K. (2016). Perceptions of climate change imagery. Science Communication, 38(2), 197–227. https://doi.org/10.1177/1075547016635181; Wong-Parodi, G., & Feygina, I. (2021). Engaging people on climate change: The role of emotional responses. Environmental	Thank you very much for these excellent and highly relevant literature suggestions. We greatly appreciate the time you dedicated to the review. We have read the recommended articles with great interest and incorporated several of them into the revised manuscript. These additions have helped us better situate our contribution within the broader climate communication and visual persuasion literature.

Communication, 15(5), 571–593.
<https://doi.org/10.1080/17524032.2020.1871051>;

Wang, S., Corner, A., Chapman, D., & Markowitz, E. (2018). Public engagement with climate imagery in a changing digital landscape. *WIREs Climate Change*, 9(2).

<https://doi.org/10.1002/wcc.509> for psychological distance place refer to this: Van Lange, P. A. M., & Huckelba, A. L. (2021).

Psychological distance: How to make climate change less abstract and closer to the self. *Current Opinion in Psychology*, 42, 49–53.
<https://doi.org/10.1016/j.copsyc.2021.03.011> and this Singh, A. S.,

Zwickle, A., Bruskotter, J. T., & Wilson, R. (2017). The perceived psychological distance of climate change impacts and its influence on support for adaptation policy. *Environmental Science & Policy*, 73, 93–99.
<https://doi.org/10.1016/j.envsci.2017.04.011>

Metag, J., Schäfer, M. S., Füchslin, T., Barsuhn, T., & Kleinen-von Königslöw, K. (2016). Perceptions of climate change imagery. *Science Communication*, 38(2), 197–227.
<https://doi.org/10.1177/1075547016635181> and this one on concrete vs. abstract images.

Duan, R., Takahashi, B., & Zwickle, A. (2021). How effective are concrete and abstract climate change images? The moderating role of construal level in climate change visual communication. *Science Communication*, 43(3), 358–387.
<https://doi.org/10.1177/10755470211008192> and this on on psychological distance:

Brügger, A. (2020). Understanding the psychological distance of climate change: The limitations of construal level theory and suggestions for alternative theoretical perspectives. *Global Environmental Change*, 60, 102023.
<https://doi.org/10.1016/j.gloenvcha.2019.102023>. this one on local framing and place attachment.

General Remarks

We would like to thank the editors, Dr. Judge and Dr. Benetreau, for the opportunity to revise and resubmit our manuscript to *Communications Sustainability*. We are also sincerely grateful to both reviewers for their careful reading of our manuscript and for their thoughtful and constructive feedback.

We have thoroughly revised the manuscript in line with the editors' and reviewers' suggestions.

In the following, we provide a point-by-point response to all comments and detail how we have addressed them. All substantial changes in the manuscript are highlighted in yellow.

Editors	Our Response
Your revised manuscript titled "Unintended Effects of AI-Generated Imagery: Realistic Climate Disaster Images Reduce Support for Climate Action" has now been seen by the same 2 reviewers, and we include their comments at the end of this message. While both reviewers appreciated the improvements made in the last revision, they still have some remaining concerns to be addressed. In particular, both reviewers note the need for the interpretations to be more closely tied to the findings. We therefore invite you to revise and resubmit your manuscript again, along with a point-by-point response that takes into account the points raised. Please highlight all changes in the manuscript text file.	We are pleased with the positive decision and grateful for the opportunity to revise and resubmit our manuscript to Communications Sustainability. We have made a considerable effort to address the reviewers' comments and to ensure that our interpretations more closely align with the observed effects. To this end, we have revised the title, abstract, introduction, and discussion sections.
Reviewer 1	Our Response
I appreciate the authors' efforts in revising the manuscript and conducting an additional replication study. I thank them for taking my concerns seriously and I can now more clearly see a path towards publication of the paper. However, I still have several substantive concerns that I think should be address before publication.	We would like to sincerely thank you for your encouraging words and for the nuanced, helpful guidance you have provided to strengthen the manuscript. Below, we detail how we have addressed your comments.

Major concerns:

1. The Paper Overstates the "Backfire" Effect

Now that the authors have conducted an additional replication study, to me, the manuscript's central claim -- that highly realistic AI disaster images reduce support for climate action -- is not consistently supported and reflects a misleading emphasis on SEM path coefficients over actual total effects. The ANCOVA results tell a different story:

Study 1: High realism vs. text-only, $p = .208$ -- this is null effect but NOT a backfire. I do note that the low realism condition moves the needle in the positive direction ($p = .044$), which is very interesting; but this is not evidence for a backfire effect.

Study 2: No significant effect of condition on willingness ($p = .156$). Again, here there is no backfire.

Study 3: A backfire emerges only when participants spontaneously suspected AI origin ($p < .001$ vs. text-only). The explicitly labeled condition does not reach significance ($p = .076$ — see the minor concern section).

Thus, here, the accurate summary is: High-realism AI disaster images consistently fail to increase willingness, but do not reliably reduce it. In Study 1, the low realism condition moves the willingness needle, which is an interesting result. Backfire is a conditional effect specific to Study 3, and only among participants who detected AI origin unprompted. The current presentation of the paper obscures this by foregrounding significant SEM indirect paths as evidence of

We thank you for your careful reading of our manuscript and for your insightful comments on this important aspect of interpretation.

While our initial understanding of the backfire effect was somewhat broader – encompassing not only overall (total) effects of AI-generated images on support for climate action, but also the presence of counter-persuasive responses such as reactance – we appreciate your point regarding the need to more closely align our interpretations with the empirical findings.

We have therefore thoroughly revised the abstract, introduction, and discussion sections. We now more clearly emphasize that AI-generated disaster images generally fail to consistently increase persuasion and that, under specific conditions (i.e., in Study 3 among participants who suspected an undisclosed AI origin), they can even decrease support for climate action.

Moreover, we have revised the title. In doing so, we carefully considered whether to foreground the overall failure of AI-generated climate disaster imagery to consistently increase support for climate action or the conditional backfire effect.

We ultimately chose to highlight the conditional backfire effect, as it represents a theoretically and practically consequential finding that may be of particular interest to the journal's readership, especially given that journals often emphasize novel or surprising results.

At the same time, we ensure that the broader pattern of findings, namely the overall failure of AI-generated climate disaster imagery to increase support for climate action, along with the underlying competing psychological mechanisms, is clearly communicated and contextualized in the abstract and throughout the manuscript.

We hope this approach is appropriate, and we would of course be open to alternative suggestions, as we recognize that title conventions and editorial preferences may vary.

backfire, while treating the null ANCOVA results as secondary. Significant indirect paths are not the same as a significant total effect. The title, abstract, and discussion should be revised to reflect that the primary finding is a failure to persuade rather than active backfire.	
2. The authors should clarify that images are not photorealistic by contemporary standards I appreciate the authors clarification of the "high realism" images. However, it should be noted that the "high realism" images used in the Experiments remain visibly stylized — dramatic lighting, saturated color grading, and an aesthetic characteristic of AI art generation circa 2022-2023. These are not photorealistic in the sense that contemporary audiences would mistake for photographs. Modern AI systems are now capable of producing genuinely photo-indistinguishable imagery. The manuscript warns against "highly realistic" AI imagery, but the stimuli used here would likely be immediately detectable as AI-generated today. All applied recommendations must be qualified accordingly: these findings apply to stylized-yet-detailed AI imagery that is detectable as AI-generated, and may not generalize to truly photorealistic outputs. I think the introduction and discussions should highlight this important caveat so that the paper findings are more readily useful for practitioners.	We fully concur that generative AI applications have made considerable progress in producing photorealistic imagery, and that the images used in our studies may not fully reflect the most recent advances in AI-generated photorealism. Nonetheless, we would like to highlight that, as reported in Study 3, a substantial number of participants perceived the images as actual photographs (N = 148). Moreover, as indicated by our pretests (see Appendix Table A.4), the highly realistic AI-generated images were consistently rated as such, with all images scoring well above the midpoint on the perceived visual realism scale. At the same time, we appreciate your point and agree that precision is important, especially in a rapidly evolving domain such as generative visual AI. We have therefore revised the introduction, methodological descriptions, and discussion section accordingly. In addition, we have added a paragraph to the discussion that explicitly reflects on the detailed, yet stylized nature of our AI-generated imagery, as well as the blurred boundaries between such images and highly edited or stylized photographs.

3. Title and Abstract Must Reflect Actual Scope and Findings The paper is specifically about AI-generated climate *disaster* imagery, and the backfire effect is conditional on perceived AI origin. Neither the title nor the abstract reflects this adequately. The paper should also clarify throughout the paper that the findings are specifically about AI-generated climate disaster images only (and not about AI-generated imagery for all kinds of climate communication) and the abstract should also clarify that backfire is conditional, not general.	We acknowledge this important point that you have raised. We have revised the title as well as the abstract accordingly to clarify the scope and streamline the interpretation of the findings. With regard to the title, we carefully considered whether to foreground the overall failure of AI-generated climate disaster imagery to consistently increase support for climate action or the conditional backfire effect. We ultimately chose to highlight the conditional backfire effect, as it represents a theoretically and practically consequential finding that may be of particular interest to the journal's readership, particularly given that journals often emphasize novel or surprising results. At the same time, we ensure that the broader pattern of findings, namely the overall failure of AI-generated climate disaster imagery to increase support for climate action, along with the underlying competing psychological mechanisms, is clearly communicated and contextualized in the abstract and throughout the manuscript. We hope this approach appropriately addresses your concern, and we would be open to alternative suggestions, as we recognize that title conventions and editorial preferences may vary. Moreover, throughout the manuscript, we have made it more clear and explicit that our findings pertain specifically to AI-generated climate disaster imagery, rather than to AI-generated imagery or climate communication more broadly.
Other Concerns: 1. Causal Language in the Mediation Claims The authors describe their central finding as: "highly realistic AI-generated disaster imagery intensifies emotional responses while simultaneously increasing perceived freedom threat and psychological reactance, which in turn reduce	Thank you for raising this important point. We have revised the respective passages to replace causal language with associative language.

willingness to make personal sacrifices for climate action." This implies a causal chain the data cannot support. Random assignment permits causal inference for the manipulation's effects, but the mediation pathways in the SEM are correlational. The authors should replace causal language ("which in turn reduce") with associative language ("which were in turn correlated with lower willingness") throughout.	
2. "Marginally Significant" Language The authors describe the AI origin label condition vs. text-only in Study 3 as producing a "marginally significant negative effect" ($p = .076$). In my opinion, this framing should be removed. $p = .076$ is not significant, and "marginal significance" is not an accepted statistical concept (if the authors want to establish this point, then they should replicate this). This should be reported as a null result.	We agree with your assessment and are therefore now treating this result as non-significant.
Summary The research question is important and the three-study structure is a strength. However, the paper currently overstates its findings through causal language, selective emphasis on SEM paths over null total effects, and insufficiently qualified claims about photorealism and climate communication in general. These issues affect the paper's central claims and must be addressed before publication.	We sincerely thank you for your careful reading and the time you dedicated to this review. We have carefully considered your comments and have made the indicated revisions, including replacing causal with associative language, streamlining and clarifying the presentation of our findings, and more precisely qualifying our claims regarding photorealism and the scope of our findings with respect to AI-generated climate disaster imagery.

Reviewer 2	Our Response
The revision substantially improves the manuscript. In particular, the added replication study, clarified terminology around visual realism, expanded theoretical framing, and fuller reporting of measures and SEM structure address several important reviewer concerns. However, some issues remain only partially resolved. The evidence for downstream persuasive “backfire” effects is still modest and somewhat uneven across studies, the concern about lack of preregistration is not fully addressed, ecological validity is acknowledged more than empirically resolved, and the methods for generating the AI images remain insufficiently documented for full reproducibility. I therefore view the manuscript as significantly strengthened, but still in need of greater methodological transparency and slightly more cautious interpretation of the findings.	We are grateful for your feedback and for seeing value in our work. We also greatly appreciate the literature sources you kindly suggested, which proved very helpful in strengthening the manuscript. We have taken your comments to heart and have made substantial efforts to address your concerns regarding the methodological transparency of the manuscript, as well as to provide a more nuanced and cautious interpretation of the data. Below, we detail how we have addressed your comments.
In particular, while the revision improves the conceptual framing, the manuscript still lacks sufficient transparency about stimulus generation. Because the core manipulation depends on AI-generated imagery, the image-production pipeline is part of the method and should be documented in much greater detail. The authors should report the exact prompts used to generate each stimulus set, preferably in an appendix or repository, along with any iterative prompt refinement, selection criteria, relevant generation settings, and whether any post-processing occurred. The manuscript should also explain why Midjourney was used for some images and ChatGPT for others, and specify exactly which ChatGPT image model/version was	Thank you for raising this important point. Following your suggestion, we have added a detailed section to the Appendix (see pp. 42 ff.), in which we document the full stimulus generation process. Specifically, we report the exact prompts used to generate all visual materials, along with information on iterative prompt refinements and the rationale underlying these modifications, generation settings, and any post-processing steps. In addition, in the Methods section of the manuscript, we clarify our use of different generative AI tools, including why Midjourney was used for certain images and ChatGPT for others, and we specify the exact ChatGPT image model/version employed. We also disclose relevant access conditions (e.g., paid subscriptions), where applicable, to enhance transparency and reproducibility.

used. Relatedly, if access depended on a paid subscription, this should be disclosed as part of the methods, since it affects reproducibility.	
Finally, the framing of AI image generation as low-cost or free should be qualified. Even if it is financially inexpensive relative to traditional production methods, GenAI image generation is not costless. It carries environmental impacts associated with inference-time computation, including energy and water use. A brief acknowledgment of this limitation would improve the paper's balance and accuracy. This is particularly relevant as it is about climate change...	You raise an important aspect that we agree we had not sufficiently addressed so far. We have therefore added a paragraph to the discussion section (p. 18), in which we explicitly address the environmental costs of generative AI, drawing on the literature you suggested as well as additional sources. Based on these references, we also derive practical implications regarding the responsible use of generative AI in visualizing the detrimental impacts of climate change. We hope that these revisions address your concern.